# *WildfireDB*: An Open-Source Dataset Connecting Wildfire Spread with Relevant Determinants

**Samriddhi Singla**[1], **Ayan Mukhopadhyay**[*2], **Michael Wilbur**[2], **Tina Diao**[3],
**Vinayak Gajjewar**[1], **Ahmed Eldawy**[1], **Mykel Kochenderfer**[3], **Ross Shachter**[3], **Abhishek Dubey**[2]
University of California, Riverside[1]
Vanderbilt University, Nashville[2]
Stanford University, Stanford[3]
{ssing068,vgajj002,eldawy}@ucr.edu[1]
{ayan.mukhopadhyay*,michael.p.wilbur,abhishek.dubey}@vanderbilt.edu[2]
{tdiao,mykel,shachter}@stanford.edu[3]

## Abstract

Modeling fire spread is critical in fire risk management. Creating data-driven models to forecast spread remains challenging due to the lack of comprehensive data sources that relate fires with relevant covariates. We present the first comprehensive and open-source dataset that relates historical fire data with relevant covariates such as weather, vegetation, and topography. Our dataset, named *WildfireDB*, contains over 17 million data points that capture how fires spread in the continental USA in the last decade. In this paper, we describe the algorithmic approach used to create and integrate the data, describe the dataset, and present benchmark results regarding data-driven models that can be learned to forecast the spread of wildfires.

## 1 Introduction

Wildfires cause loss of life, economic damage, and pose indirect environmental and health threats [Doerr and Santín, 2016]. The November 2018 Camp Fire in Northern California resulted in losses worth $24 billion, including property destruction and firefighting costs [Bartz, 2019]. Occurrences of such extreme fire events are likely to increase [Joseph et al., 2019]. In 2021, a total of 40,945 wildfires occurred across the United States of America that burned over 4.4 million acres (as of August) [Center for Disaster Philanthropy, 2021]. The effects of wildfires are devastating; power outages, forced evacuation, destruction of housing and business, loss of life, and long-term environmental consequences make it imperative that principled and accurate methods are developed to forecast the spread of wildfires and respond to them.

Modeling the dynamics of fire spread is crucial to first responders. Responders need to allocate limited resources across large areas to combat fires and minimize the loss of life and property [Diao et al., 2020]. Traditionally, fire spread is modeled by tools that use *physics-based* modeling [Rothermel, 1972, Andrews, 1986, Finney, 1998]. While such models are widely used, the prediction of fire spread can be improved by a large set of covariates. However, it is difficult to model the exact effect of each covariate on the spread of fire in closed-form. Data-driven modeling can be used to estimate the effects of a diverse set of features on wildfire susceptibility (such as geographic and climate data) [Joseph et al., 2019, Ghorbanzadeh et al., 2019] and improve response to emergency incidents in general [Mukhopadhyay et al., 2020]. However, to the best of our knowledge, there is no comprehensive and open-source data source that combines fire occurrences with topography, vegetation, and weather to allow the research community to develop approaches to manage wildfires.

---

*Corresponding Author

35th Conference on Neural Information Processing Systems (NeurIPS 2021) Track on Datasets and Benchmarks.

Generating a comprehensive dataset about wildfires is difficult due to many reasons. **First**, data regarding fire occurrence and covariates are often collected and stored by different agencies and sources. Also, such data are usually available in different data models. For example, the locations and sizes of historical fire occurrences are usually available in a vector model, while information about vegetation, fuel, and topographic features is available in a raster model. These two data models use different storage mechanisms and computational methods that make it difficult to combine them. **Second**, fires spread through extremely large areas through which covariates can vary significantly. As an example, the number of spatial units we use to gather data about vegetation type (and other features) is over 18 billion. Mining such large-scale feature data is a massive computational bottleneck. The large size of the data sources further complicates the fusion of raster and vector data. **Third**, while the weather is an important determinant of fire spread, obtaining granular weather data is challenging.

Through this paper, we make available a spatio-temporal dataset, *WildfireDB*, that can be used to model how wildfires spread as a function of relevant covariates. Our data generation, to a large extent, is based on our previous work on large-scale vector and raster analysis [Singla et al., 2021], which introduces a novel algorithmic approach to merge large scale raster and vector data. To create a comprehensive dataset regarding wildfire propagation, we use the area of continental United States, covering a total area of 8,080,464 sq. kilometers. We look at every wildfire occurrence in our area of interest from 2012-2017. Since the spread of wildfires is largely determined by determinants such as weather, topography, and vegetation cover, we extract relevant information from other data streams and merge them with information about fire occurrences. Our dataset consists of a total of 17,820,835 data points.

The rest of the paper is organized as follows. We first describe the data sources used to create the dataset in section 2. Then, we describe the algorithmic approach used to generate the data in section 3. Benchmark results and intended use of the dataset is presented in section 5 and section 6 respectively. We follow the framework described by Gebru et al. in "datasheet for datasets" [Gebru et al., 2018] to create a detailed documentation for the proposed dataset, which is presented in the appendix.

## 2  Data Sources

Our goal is to present a data source that can be used to model both fire occurrences and the spread of fires. We discretize space and time to create our dataset (we discuss the unit of discretization below). Each entry in the dataset consists of a specific spatial location (we refer to such spatial units as *cells*) that is observed to be on fire at a particular time-step along with spatially-associated vegetation descriptors, weather parameters, and topography information. Each entry also consists of fire occurrence and the same set of features in a neighboring cell at the subsequent time-step.

The fire occurrence data is collected in vector form from the Visible Infrared Imaging Radiometer Suite (VIIRS) thermal anomalies/active fire database [Schroeder et al., 2014, NASA, 2020]. The database provides information from the VIIRS sensors on the joint NASA/NOAA Suomi National Polar-orbiting Partnership (Suomi NPP) and NOAA-20 satellites. While it is also possible to use Moderate Resolution Imaging Spectroradiometer (MODIS) fire detection, we choose VIIRS sensor data due to its improved spatial resolution and night-time performance. The dataset contains latitude and longitude values corresponding to the center of pixels representing $375 \times 375$-meter square cells. An incidence of fire is indicated by the fire radiative power (FRP) of the concerned pixel (cell), which corresponds to the pixel-integrated fire radiative power in megawatts. Due to the increased spatial and spectral resolution of the data released by NASA, their fire detection algorithm is tuned to optimize its performance even for small fires while balancing the occurrence of false positives [NASA, 2020]. The temporal granularity of the data depends on the specific area of interest and the frequency with which satellites sweep over the area. For the continental United States, we observe that the temporal frequency of fire detection varies between 1 and 8 observations per day, with about $85\%$ of the area under consideration having 1 observation per day. As a result, we use a temporal discretization of a day. However, we report all observations for spatial locations that have more than a single observation in a day.

The vegetation and topography data are collected in raster form from the "LANDFIRE" project [Ryan and Opperman, 2013], which is based on satellite imagery. The raster files have a spatial resolution of $30 \times 30$ meter square cells. For each feature, such as vegetation cover, we process approximately

Table 1: LANDFIRE raster data categories

| Name | Year(s) |
|---|---|
| Canopy Base Density | 2012, 2014, 2016 |
| Canopy Base Height | 2012, 2014, 2016 |
| Canopy Cover | 2012, 2014, 2016 |
| Canopy Height | 2012, 2014, 2016 |
| Existing Vegetation Cover | 2012, 2014, 2016 |
| Existing Vegetation Height | 2012, 2014, 2016 |
| Existing Vegetation Type | 2012, 2014, 2016 |
| Elevation | 2016 |
| Slope | 2016 |

15 billion raster pixels. We perform such data processing for data categories such as canopy base density, canopy cover, and vegetation type. We list all the data categories gathered from the Landfire project in Table 1. Further, we collect weather data from Meteostat [Meteostat, 2020], an online service that provides weather and climate statistics around the globe. Meteostat collects raw data from the National Oceanic and Atmospheric Administration (NOAA) for the continental USA. We gather aggregated daily weather data for 5,787 weather stations. Weather parameters include average, minimum, and maximum temperature, total precipitation, average atmospheric pressure, average wind speed, and average wind direction.

## 3   Data Generation

To reconcile the different spatial resolutions, we divide the spatial area under consideration into a grid $G$ consisting of $375 \times 375$ meter cells (the resolution of VIIRS data), resulting in over 55 million polygons. Let an arbitrary cell be denoted by $g_i \in G$. The center of each fire pixel from the vector data can therefore overlap with exactly one cell in the grid. To compute the corresponding vegetation and topographic information associated with each data point, we compute *zonal statistics* for the vector data using the raster data. The method of zonal statistics calculation refers to computing summary statistics using a raster dataset within zones defined by another dataset (typically in vector form). We show the process in Fig. 1.

The data generation process as depicted in Figure 2 includes the following steps: **1.** Compute zonal statistics for each spatial cell (in the form of a polygon in the vector data) using the LANDFIRE raster data. **2.** Find the spatial neighbors for each cell. We define spatial neighborhood of a cell as the set of all adjacent cells in the grid. **3.** For each fire observed in the VIIRS dataset, find its corresponding cell in the grid. **4.** Let an arbitrary observation denoting a fire be represented by $(g_i, t)$ where $g_i \in G$ and $t$ denote the location of the fire and its observed time of occurrence respectively. For each such point, we generate the tuple $\{g_i, t, f_i, x_{it}, g_n, t+1, f_n, x_{n(t+1)}\}$, where $g_i$ and $g_n$ are neighbors, $g_i$ is observed to be on fire at time-step $t$, and $g_n$ may or may not be burning at time-step $t+1$. The feature vectors for the cells $g_i$ and $g_n$ are denoted by $f_i$ and $f_n$ respectively. Similarly, $x_{it}$ and $x_{n(t+1)}$ denote the FRP value of cell $g_i$ (at time $t$) and cell $g_n$ (at time $t+1$) respectively. **5.** For each tuple generated in the last step, find its corresponding weather data $W$ and append it to the tuple. We describe each step below.

**1.  Compute Zonal Statistics:** For each spatial cell in the 375m $\times$ 375m grid placed over the continental USA and for each raster dataset mentioned in Table 1, we want to compute aggregated feature vectors. To compute zonal statistics, we employ the fully distributed system, *Raptor Join* proposed in Singla et al. [2021] on an Amazon AWS EMR cluster with one head node and 19 worker nodes of type m4.2xlarge with 2.4 GHz Intel Xeon $E5 - 2676$ v3 processor, 32 GB of RAM, up to 100 GB of SSD, and 2$\times$8-core processors. We work with data in their native formats by computing an intermediate data structure called *intersection file* that maps raster to vector data. The creation of *intersection file* also facilitates the use of distributed computing to compute zonal statistics, thereby enabling the processing of massive vector and raster data. This process outputs a collection of tuples $(g_i, Geometry_i, f_i)$ where $Geometry_i$ and $f_i$ refer to the actual spatial geometry and the set of feature values calculated for cell $g_i$ respectively.

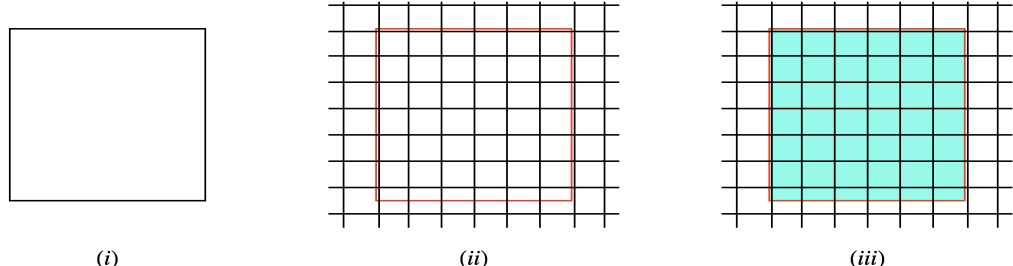

Figure 1: Calculating zonal statistics. (i) An example cell $g_i \in G$ from VIIRS fire occurrence data with a spatial resolution of 375m x 375m. (ii) Multiple LANDFIRE pixels (available in the form of raster data) overlap with $g_i$. (iii) The process of calculating zonal statistics computes the set of raster pixels that overlap with $g_i$ (shown in blue) and then calculates summary statistics using the set

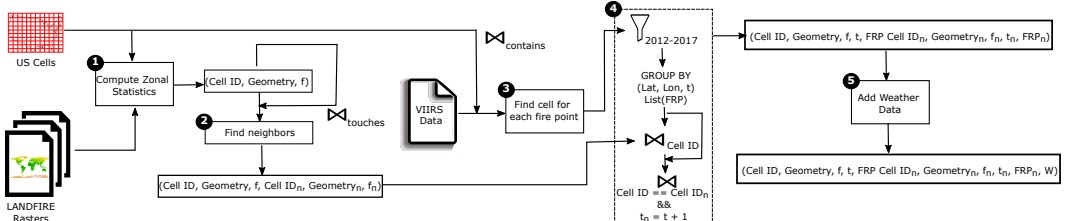

Figure 2: Data Generation Process

**2. Find neighbors:** The neighbors for each cell in the spatial grid are computed by doing a spatial self-join using the predicate *touches* on the $Geometry$ values of the tuples generated in the previous step. The predicate *touches* returns true, if and only if the boundaries of the cells intersect. We implement the spatial join using Beast [Eldawy et al., 2021]. It outputs a collection of tuples $(g_i, Geometry_i, f_i, g_n, Geometry_n, f_n)$ where each tuple in the previous step is appended by the tuples of one of its neighbors (recall that we use subscript $n$ to denote variables that refer to the neighbors of the cell in consideration).

**3. Find cell for each fire point:** For specific points (latitude-longitude pairs) in VIIRS data and the cells in our spatial grid, a spatial join using the predicate *contains* is performed to find the cell that each fire point is contained in. The predicate *contains* returns true, if and only if the fire point lies in the interior of the cell. This step is implemented using Beast [Eldawy et al., 2021].

**4. Generate tuples:** To generate the final tuples for *WildfireDB*, we start by filtering the tuples in the VIIRS data for the years 2012 to 2017. The VIIRS dataset may contain multiple tuples for the same fire point having the same time-step yet different FRP values. We group all such tuples by the fire point and time-step and create a list for the FRP values to generate a single tuple. The resulting VIIRS tuples are then joined with tuples from Step 2. Finally, we filter information about each neighbor of the cell under consideration at the next time step. This results in tuples of the form $(g_i, Geometry_i, f_i, t, x_{it}, g_n, Geometry_n, f_n, t+1, x_{n(t+1)})$. If the condition on the neighbor's time-step is not satisfied, the value of $x_{n(t+1)}$ is set to zero, i.e. no fire.

**5. Weather:** To incorporate weather, for each tuple generated in step 4, we find all weather stations within 160 km and sort the stations by distance to the centroid of the cell's geometry. For each weather parameter, we find the weather reading from the nearest weather station for which a valid reading was available. If there is no valid weather reading for a given parameter from a station within 160 km of the centroid of a cell on a given day, a null value is recorded for that parameter. Recall that each data point in our dataset consists of information about two spatial cells: a reference cell observed to be on fire at a time-step and a neighboring cell that may or may not be on fire in the subsequent time-step. Since we want to capture how fire spreads spatially, we incorporate how the wind blows relative to the reference and the neighboring cell. Consider an arbitrary tuple with reference cell $g_i \in G$ and neighboring cell $g_j \in G$. Let the time under consideration be denoted by $t$. Let the average wind speed be $w_s$ and the average meteorological wind direction be $\varphi_w$ at time $t$. We follow the convention of measuring meteorological wind direction with respect to the north-south line in a

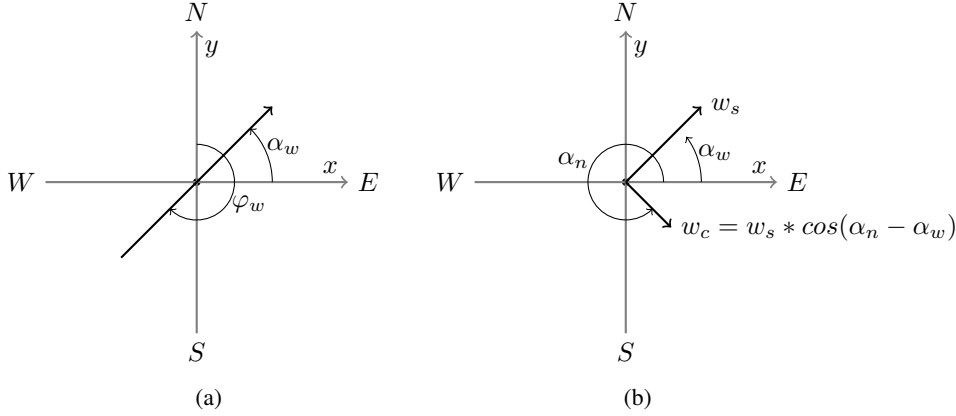

(a)                                     (b)

Figure 3: (a) A SW wind (southwesterly) with meteorological wind direction $\varphi_w = 225°$, which corresponds to a polar wind direction of $\alpha_w = 45°$. (b) Calculating the wind component ($w_c$) of a southwesterly wind with $\alpha_w = 45°$ and magnitude of $w_s$ on a neighboring cell directly southeast ($\alpha_n = 315°$).

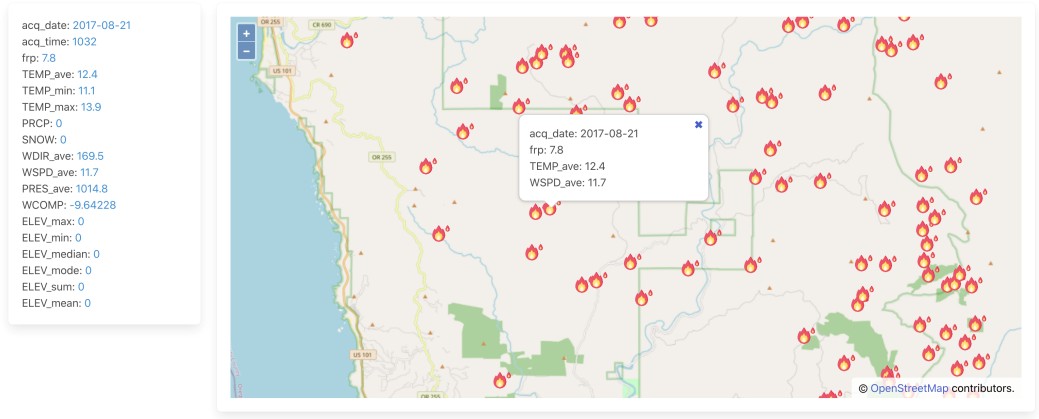

Figure 4: WildfireDB Visualization Interface

clockwise manner (see Fig. 3). Let the component of $w_s$ that lies on the line connecting $g_i$ and $g_j$ be denoted by $w_c$. The component $w_c$ can be calculated as $w_s \cos(\alpha_n - \alpha_w)$, where $\alpha_n = \text{atan2}(y, x)$, where $y$ and $x$ are the difference in latitude and longitude between the cells respectively (see Figure 3 for details).

## 4    Data Availability

In order to enable practitioners and researchers access the data based on their needs, we created an online interface where users can visualize the WildfireDB dataset. The users can use pan and zoom functions to explore the data. Users can also download data based on a specific area of interest or by using a specific zoom level (or download the entire data at once). Information pertaining to individual fires can also be seen by hovering or clicking. We show the interface in Figure 4. The interface can be accessed through `raptor.cs.ucr.edu/wildfiredb`. We maintain an up-to-date description about the data at `https://wildfire-modeling.github.io/`. The data itself is hosted at `https://doi.org/10.5281/zenodo.5636429`.

# 5 Benchmark Results

While our goal is to release an open-source dataset that links fire occurrence with relevant determinants, we also present some results using standard learning approaches to show how *WildfireDB* can be used to forecast the spread of wildfires. The results are presented using data instances that did not have any missing values; we do not perform (or suggest) any data imputation to preserve the original data that was collected. Practitioners and users can choose to replace missing values based on the context and domain expertise. The total number of data points we use for learning is 9,210,537. We use 80% of the data for training and 20% for testing, resulting in 7,368,430 data points in the train set and 1,842,107 in the test dataset.

We present results using three regression models, namely linear regression, random forests, and artificial neural network (ANN) [Murphy, 2012, Goodfellow et al., 2016]. The results of the regression models are provided in Table 2. The random forest model consists of a maximum depth of 30, maximum samples per iteration of 5,000, and trees are varied between 50, 100, and 500. We find that the random forest model was insensitive to the number of trees. The neural network regression model consists of four hidden layers of 500, 400, 128, and 64 units respectively with ReLU activation and a single neuron in the output layer with linear activation. We use the Adam optimizer [Kingma and Ba, 2014] with a learning rate of 0.001 to train the network. We find that all the regression models have similar performance.

Forecasting fire intensity can also be presented as a classification problem. In this case, a forecast is labeled as a true positive when both the predicted fire intensity and the recorded fire intensity are greater than a pre-specified threshold ($\epsilon$, say). We investigate three threshold values (0.5, 1.0, and 5.0). The results of the classification models (random forest, logistic regression, and neural network) are provided in Table 3. We observe that both regression and classification models perform poorly when trained using *WildfireDB*. We hypothesize that this result is primarily due to the extremely high sparsity in the dataset. We point out that our goal in this paper is solely to create, document, and explain the data source and not focus on modeling fire spread. As a result, we only provide benchmark results using standard approaches and do not seek to create models that can outperform such baselines.

Table 2: Regression Models

| Model | MSE | MAE |
|-------|-----|-----|
| Linear Regression | 45.88 | 1.08 |
| Random Forest | 46.53 | 1.09 |
| Neural Network | 46.73 | 1.08 |

Table 3: Classification Models

| Model | FRP Threshold ($\epsilon$) | Accuracy | Precision | Recall |
|-------|---------------------------|----------|-----------|--------|
| Random Forest | 0.5 | 93% | 0.62 | 0.00 |
| Random Forest | 1.0 | 94% | 0.61 | 0.00 |
| Random Forest | 5.0 | 98% | 0.00 | 0.00 |
| Logistic Regression | 0.5 | 93% | 0.31 | 0.02 |
| Logistic Regression | 1.0 | 94% | 0.26 | 0.01 |
| Logistic Regression | 5.0 | 98% | 0.00 | 0.00 |
| Neural Network | 0.5 | 93% | 0.00 | 0.00 |
| Neural Network | 1.0 | 94% | 0.00 | 0.00 |
| Neural Network | 5.0 | 98% | 0.00 | 0.00 |

# 6 Discussion

**Intended Use**: As we show, the proposed dataset can be used to model the spread of wildfires as a function of relevant determinants like vegetation, altitude, canopy height, and weather. We hypothesize that the dataset can be used to achieve the following: a) simulate how fires can spread

in real-time to assist evacuation, b) inform agent-based modeling for planning emergency response, c) simulate how fires might spread in the near future under the effects of deforestation and climate change, and d) aid planning of new infrastructure development and analyze risk from potential fires. Although the dataset is primarily designed to capture the spread of historic fires, it implicitly also contains data about fire occurrence itself. For example, it is possible to isolate spatial cells where fires originated by searching the spatial-temporal neighborhood of the fires. Using such an analysis, it is then possible to model the occurrence of fires.

**Limitations**: We point out the following limitations and words of caution about using the data. **1)** Although the dataset captures relevant determinants of fire spread, note that we do not claim any causal link between the features and the spread of fire. For example, consider that a cell $g_i \in G$ is observed to be on fire at time-step $t$ and a neighboring cell $g_j \in G$ is observed to be on fire at the subsequent time step $t + 1$. Although such data can be used to model the spread of fire, we do not claim (for reasons mentioned below) that the fire indeed spread to $g_j$ from $g_i$. The fire could have spread to $g_j$ from a different neighbor (which would also be captured by the data), or a new fire could have started in it. **2)** Since wildfires pose threats to human lives, infrastructure, and the environment, first responders and firefighters respond to wildfires in order to suppress their spread. Such intervention focuses typically by targeting one of the two important ingredients that help fires spread – fuel and heat. Firefighters reduce heat by using water or fire retardant either on the ground or through the air through airplanes. Fuel is removed by clearing areas of vegetation, which can include planned fires [US Department of Interior, 2020]. Our dataset does not consist of any information about such interventions. For example, consider a cell observed to be on fire through satellite imagery. Firefighters can respond to the fire before it spreads to a neighboring cell; while the fire *not* spreading to the neighboring cell is captured by our data, the intervention by firefighters is not captured. **3)** Parts of the data we use to aggregate *WildfireDB* is generated through modeling on raw data. For example, the VIIRS fire product [NASA [2020]] determines the presence of fire by balancing the detection of small fires while simultaneously minimizing false positives. Any noise or inaccuracies generated through such modeling is also present in the proposed dataset.

# 7  Conclusion

We present an open-source dataset that captures fire occurrences in continental United States between 2012-17. We also extract information about determinants of fire spread such as topography, vegetation, and weather. Our data, presented in a discretized spatial and temporal setting, can be used to model fire spread as a function of different covariates. We also present benchmark results using well-known machine learning algorithms. Importantly, our findings reveal that it is difficult to model the spread of fire using machine learning models due to high sparsity in the data. To the best of our knowledge, *WildfireDB* is the first open-source dataset that connects occurrences of wildfires with topography, vegetation, and weather.

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
