# OpenReview forum: "WildfireDB: An Open-Source Dataset Connecting Wildfire Occurrence with Relevant Determinants"
_NeurIPS.cc/2021/Track/Datasets_and_Benchmarks/Round2 — NeurIPS 2021 Datasets and Benchmarks Track (Round 2)_

### Official Review · Reviewer_w9ep · 2021-09-08
**Despite the large scale, the data generation is not novel**

**Rating:** 4
**Confidence:** 4
**Clarity:** The paper is clearly written.

**Strengths:**

1. The dataset is of significantly large scale (consisting of 17M data points and covering 8M sq. kilometers).
2. Many determinants such as weather, topography, and fuel levels are provided and merged with the information about fire occurrences.
3. Baseline methods like linear regression, random forest, and neural network are evaluated and analyzed.

**Weaknesses:**

1. The data generation, to a large extent, is based on their previous work on large-scale vector and raster analysis [Singla and Eldawy, 2020].
2. The problem of wildfire occurrence is not expected to draw much attention in academia.
3. Only the very basic baselines are studied. Some state-of-the-art approaches should be included.

**Additional Feedback:**

1. For the evaluation with neural network, it may be useful to compare different backbone architectures.
2. Please compare with other open-source datasets for similar problems.
3. Will there be an official leader board maintained by the creators of the dataset?

**Correctness:**

The dataset is constructed in a sound way. The evaluation of baseline methods is designed in an appropriate manner.

**Documentation:**

The dataset is available and maintained at https://wildfire-modeling.github.io/.

**Ethics:**

There is no known ethical issue.

**Relation To Prior Work:**

The data generation is based on their previous work on large-scale vector and raster analysis [Singla and Eldawy, 2020].

**Summary And Contributions:**

In this work, an spatio-temporal dataset that models how wildfires spread as a function of relevant covariates is proposed. The data generation is based on their previous work on large-scale vector and raster analysis [Singla and Eldawy, 2020]. There are a total of 17M data points. Evaluation with baseline machine learning methods is given.

---

> ### Author Response · Authors · 2021-09-25
> **Clarification/discussion with Reviewer 3 (w9ep)**
>
> We thank the reviewer for raising important questions and providing valuable feedback.
>
> **Re: Novelty of Data generation** – We agree that the data generation is entirely based on our prior work which is a general framework for merging very large-scale vector and raster data. However, **the dataset itself is novel. There are no large-scale open-source datasets that connect wildfire occurrences with topography, fuel, and weather.** We believe that novel datasets fit well with the criteria of this DB track.
>
> **Re: Academic importance** – Actually, **wildfire prediction and response has recently drawn significant attention in academia due to extensive damage caused by wildfires to human lives, infrastructure, forests, and wildlife (including multiple large grants from NSF to investigate this problem)**. You are correct in the sense that it has probably not drawn as much attention as the scale of the problem demands. We believe this is partly due to the lack of open-source datasets in this domain. We seek to bridge this gap.
>
> **Re: Evaluations** – We purposefully kept evaluations restricted to well-known yet simple models. This is motivated by two reasons – first, our goal in this **db paper is to release an extensive open-source dataset**. Second, **learning using data exhibiting very high sparsity (our data shows that 93.1% of the fires do not spread) typically needs to leverage structure in the problem to improve prediction**, which is outside the scope of the current paper and is a novel research direction in itself.
>
> **Re: Leaderboard** – This is a great suggestion! We will definitely maintain a leaderboard!

---

### Official Review · Reviewer_3DAT · 2021-09-20
**A large dataset on wildfires in the United States combining information of geography, fires, topography, vegetation, and weather.**

**Rating:** 7
**Confidence:** 5
**Clarity:** Everything is very clear.

**Strengths:**

Several tasks were done:
- the discretization of the space;
- the integration of data from different sources;
- the creation of an online interface;
- two benchmark studies for both regression and classification using 3 different algorithms



**Weaknesses:**

- how were the hyper-parameters defined, e.g., for neural networks? This is not explained.



**Additional Feedback:**



**Correctness:**

Good
TYPOS
line 188: this results is largely -> these results are largely

**Documentation:**

Is everything clearly supported.

**Ethics:**

I believe no ethical issues

**Relation To Prior Work:**

No prior work reported

**Summary And Contributions:**

This paper presents a dataset on wildfires that integrates data from different sources which allows the inclusion of possible determinants.
It also presents a benchmark study using 3 different methods: linear regression, random forest, and neural networks. The problem is addressed both as a regression and as a classification problem.

---

> ### Author Response · Authors · 2021-09-25
> **Clarifications/discussion with Reviewer 2 (3DAT)**
>
> We thank the reviewer for the feedback. We will correct the typo and update the experiments section with details re: hyper-parameters for the neural network.

---

### Official Review · Reviewer_objf · 2021-09-21
**Wildfire DB -- focus on related environmental factors -- low model performance**

**Rating:** 4
**Confidence:** 2
**Clarity:** Paper is well written/easy to read.

**Strengths:**

Pulls in data that’s scattered across a number of different organizations. Clear/easy to read

**Weaknesses:**

Susceptibility to fire also seems like it’s a function of the underlying built structures as well as some degree of human behavior (mitigating ignition conditions). How do these factors appear in this dataset? Are they fuel? Or is that anticipated in a future version?

This might explain why the performance of the models is also somewhat low – lots of missing variables. That isn’t to say this starting point is not useful, it just seems constrained by how much may be missing/unexplained. I'm not particularly clear on the value-add here.


**Additional Feedback:**

Wildfire is extremely important and so the frame is on point. I don't understand why these are the variables that are included at the expense of others though. Are there additional people or papers who are dissatisfied with the current state of affairs so that they can cite/subtantiate this comment, which seems to be the premise of the dataset collection? "While such [physics-based ]models are widely used, prediction of fire spread can be improved by a large set of covariates"

**Correctness:**

The dataset appears to be constructed soundly and has documentation, though is not as detailed as many other submissions I’ve evaluated (missing the Gebru dataset datasheet, for example, and doesn’t describe *what* types of weather they’re pulling in).

Timnit Gebru, Jamie Morgenstern, Briana Vecchione, Jennifer Wortman Vaughan, Hanna
381 Wallach, Hal Daume´ III, and Kate Crawford. Datasheets for datasets. arXiv:1803.09010, 2018.

**Documentation:**

Has a website and github repo but not as detailed as many of the other datasets I've reviewed for this call, also unclear what the maintenance plan is.

**Relation To Prior Work:**

What about papers that discuss things like home ignition conditions? Here's another sample other work that speaks to this:

Calkin, D.E., Cohen, J.D., Finney, M.A. and Thompson, M.P., 2014. How risk management can prevent future wildfire disasters in the wildland-urban interface. Proceedings of the National Academy of Sciences, 111(2), pp.746-751.


**Summary And Contributions:**

This dataset includes over 17 million datapoints relating to fire spread in the continental US over 2012-2017 timeframe. As wildfires pose extreme risks to human lives and livelihoods, having a coherent opensource dataset that this team presents could enables a better understanding of how fires spread can help develop improved strategies to contain their spread and mitigate losses. This particular dataset focuses in on fires as detected from the VIIRS sensors as well as vegetation height, elevation, and slope from a different dataset entitled LANDFIRE (presumably from Landsat records), and daily meteorological variables (temp, windspeed) from NOAA (via Meteosat). In a sample test via classification and regression, all models performed relatively poorly.

---

> ### Author Response · Authors · 2021-09-25
> **Clarification/Discussion with Reviewer 1 (objf)**
>
> We thank the reviewer for raising important questions and providing valuable feedback.
>
> **Re: documentation** — actually, **we do provide the datasheet for datasets (from Gebru et al.) in the appendix of the paper**. Also, as mentioned on page 3, our weather parameters include average, minimum, and maximum temperature, total precipitation, average atmospheric pressure, average wind speed and average wind direction. We can add definitions of each of the features if desired.
>
> **Re: function of underlying structures** — you are correct, underlying built structures play a role in the spread of fires, especially when fires get close to inhabited areas. However, most wildfires stem from areas covered with vegetation. Also, it is extremely difficult to incorporate granular details about human made structures for extremely large geographic areas (e.g., continental USA). **As you correctly hypothesize, we approximate such factors through fuel**. We will let users fork our data and push newer versions by adding features.
>
> **Re: performance** — poor performance of the learning models is largely driven by **very high sparsity in the data (our data shows that 93.1% of the fires do not spread)**. This is a well-known problem (high precision and low recall) in sparse spatial-temporal events (https://arxiv.org/abs/2006.04200). However, our goal in this paper was to primarily introduce the dataset and baseline models. We welcome the research community to use our dataset to improve learning.
>
> **Re: maintenance** — We do mention this briefly in the appendix. Our current plan is to maintain both versions (a prior release at NeurIPS AI for Earth Sciences with 2 million data points) and the current release (with 17 millions data points). The maintenance is done by SS, AM, and MW. The database will be hosted at UCR Star, an active spatial-temporal data repository maintained by UC Riverside that hosts terabytes of public geospatial data through interactive interfaces. **We will clarify the hosting details in the paper**.
>
> **Re: learning vs physics-based models** –- This is a really interesting point. Over the last few years, learning-based models have shown improved performance in predicting fire spread (e.g., Forest Fire Spread using Deep Learning, and many others). A constraint in designing learning based approaches is the non-availability of open-source data, a gap that we seek to bridge.

---

### Decision · Program_Chairs · 2021-10-11

**Decision:**

Accept

**Comment:**

In this paper the authors present a new dataset for wildfires spread with features including weather and topography. Further, they offer relatively simple baselines to predict the spread of wildfires. While there was a wide range of perspectives from reviewers, all reviewers largely agreed with the construction of the dataset as well as the clarity and validity of the work. While there were some concerns about the breadth of study in the ML community and the need for growth of research in this area (both for even more data and more models), the paper still contributes enough for it to be valuable to the academic community. As such, and after consulting with a second AC, I recommend acceptance and also encourage the authors to expand the paper in the ways suggested by the reviewers.